# Policy Gradient With Value Function Approximation For Collective Multiagent Planning

**Duc Thien Nguyen    Akshat Kumar    Hoong Chuin Lau**
School of Information Systems
Singapore Management University
80 Stamford Road, Singapore 178902
{dtnguyen.2014,akshatkumar,hclau}@smu.edu.sg

## Abstract

Decentralized (PO)MDPs provide an expressive framework for sequential decision making in a multiagent system. Given their computational complexity, recent research has focused on tractable yet practical subclasses of Dec-POMDPs. We address such a subclass called $\mathbb{C}$Dec-POMDP where the collective behavior of a population of agents affects the joint-reward and environment dynamics. Our main contribution is an actor-critic (AC) reinforcement learning method for optimizing $\mathbb{C}$Dec-POMDP policies. Vanilla AC has slow convergence for larger problems. To address this, we show how a particular decomposition of the approximate action-value function over agents leads to effective updates, and also derive a new way to train the critic based on local reward signals. Comparisons on a synthetic benchmark and a real world taxi fleet optimization problem show that our new AC approach provides better quality solutions than previous best approaches.

## 1   Introduction

Decentralized partially observable MDPs (Dec-POMDPs) have emerged in recent years as a promising framework for multiagent collaborative sequential decision making (Bernstein et al., 2002). Dec-POMDPs model settings where agents act based on different partial observations about the environment and each other to maximize a global objective. Applications of Dec-POMDPs include coordinating planetary rovers (Becker et al., 2004b), multi-robot coordination (Amato et al., 2015) and throughput optimization in wireless network (Winstein and Balakrishnan, 2013; Pajarinen et al., 2014). However, solving Dec-POMDPs is computationally challenging, being NEXP-Hard even for 2-agent problems (Bernstein et al., 2002).

To increase scalability and application to practical problems, past research has explored restricted interactions among agents such as state transition and observation independence (Nair et al., 2005; Kumar et al., 2011, 2015), event driven interactions (Becker et al., 2004a) and weak coupling among agents (Witwicki and Durfee, 2010). Recently, a number of works have focused on settings where agent identities do not affect interactions among agents. Instead, environment dynamics are primarily driven by the *collective* influence of agents (Varakantham et al., 2014; Sonu et al., 2015; Robbel et al., 2016; Nguyen et al., 2017), similar to well known congestion games (Meyers and Schulz, 2012). Several problems in urban transportation such as taxi supply-demand matching can be modeled using such collective planning models (Varakantham et al., 2012; Nguyen et al., 2017).

In this work, we focus on the *collective* Dec-POMDP framework ($\mathbb{C}$Dec-POMDP) that formalizes such a collective multiagent sequential decision making problem under uncertainty (Nguyen et al., 2017). Nguyen et al. present a sampling based approach to optimize policies in the $\mathbb{C}$Dec-POMDP model. A key drawback of this previous approach is that policies are represented in a tabular form which scales poorly with the size of observation space of agents. Motivated by the recent suc-

cess of reinforcement learning (RL) approaches (Mnih et al., 2015; Schulman et al., 2015; Mnih et al., 2016; Foerster et al., 2016; Leibo et al., 2017), our main contribution is a actor-critic (AC) reinforcement learning method (Konda and Tsitsiklis, 2003) for optimizing $\mathbb{C}$Dec-POMDP policies.

Policies are represented using function approximator such as a neural network, thereby avoiding the scalability issues of a tabular policy. We derive the policy gradient and develop a factored action-value approximator based on collective agent interactions in $\mathbb{C}$Dec-POMDPs. Vanilla AC is slow to converge on large problems due to known issues of learning with *global* reward in large multiagent systems (Bagnell and Ng, 2005). To address this, we also develop a new way to train the critic, our action-value approximator, that effectively utilizes local value function of agents.

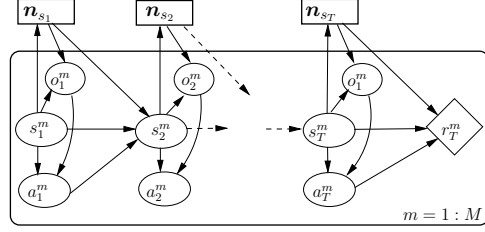

Figure 1: T-step DBN for a $\mathbb{C}$Dec-POMDP

We test our approach on a synthetic multirobot grid navigation domain from (Nguyen et al., 2017), and a real world supply-demand taxi matching problem in a large Asian city with up to 8000 taxis (or agents) showing the scalability of our approach to large multiagent systems. Empirically, our new factored actor-critic approach works better than previous best approaches providing much higher solution quality. The factored AC algorithm empirically converges much faster than the vanilla AC validating the effectiveness of our new training approach for the critic.

**Related work:** Our work is based on the framework of policy gradient with approximate value function similar to Sutton et al. (1999). However, as we empirically show, directly applying the original policy gradient from Sutton et al. (1999) into the multi-agent setting and specifically for the $\mathbb{C}$Dec-POMDP model results in a high variance solution. In this work, we show a suitable form of compatible value function approximation for $\mathbb{C}$Dec-POMDPs that results in an efficient and low variance policy gradient update. Reinforcement learning for decentralized policies has been studied earlier in Peshkin et al. (2000), Aberdeen (2006). Guestrin et al. (2002) also proposed using REINFORCE to train a softmax policy of a factored value function from the coordination graph. However in such previous works, policy gradient is estimated from the global empirical returns instead of a decomposed critic. We show in section 4 that having a decomposed critic along with an individual value function based training of this critic is important for sample-efficient learning. Our empirical results show that our proposed critic training has faster convergence than training with global empirical returns.

## 2 Collective Decentralized POMDP Model

We first describe the $\mathbb{C}$Dec-POMDP model introduced in (Nguyen et al., 2017). A $T$-step Dynamic Bayesian Network (DBN) for this model is shown using the plate notation in figure 1. It consists of the following:

- A finite planning horizon $H$.
- The number of agents $M$. An agent $m$ can be in one of the states in the state space $S$. The joint state space is $\times_{m=1}^{M} S$. We denote a single state as $i \in S$.
- A set of action $A$ for each agent $m$. We denote an individual action as $j \in A$.
- Let $(s_{1:H}, a_{1:H})^m = (s_1^m, a_1^m, s_2^m \ldots, s_H^m, a_H^m)$ denote the complete state-action trajectory of an agent $m$. We denote the state and action of agent $m$ at time $t$ using random variables $s_t^m$, $a_t^m$. Different indicator functions $\mathbb{I}_t(\cdot)$ are defined in table 1. We define the following count given the trajectory of each agent $m \in M$:

$$n_t(i, j, i') = \sum_{m=1}^{M} \mathbb{I}_t^m(i, j, i') \; \forall i, i' \in S, j \in A$$

As noted in table 1, count $n_t(i, j, i')$ denotes the number of agents in state $i$ taking action $j$ at time step $t$ and transitioning to next state $i'$; other counts, $n_t(i)$ and $n_t(i, j)$, are defined analogously. Using these counts, we can define the count tables $\mathbf{n}_{s_t}$ and $\mathbf{n}_{s_t a_t}$ for the time step $t$ as shown in table 1.

| | |
|---|---|
| $\mathbb{I}_t^m(i) \in \{0,1\}$ | if agent $m$ is at state $i$ at time $t$ or $s_t^m = i$ |
| $\mathbb{I}_t^m(i,j) \in \{0,1\}$ | if agent $m$ takes action $j$ in state $i$ at time $t$ or $(s_t^m, a_t^m) = (i,j)$ |
| $\mathbb{I}_t^m(i,j,i') \in \{0,1\}$ | if agent $m$ takes action $j$ in state $i$ at time $t$ and transitions to state $i'$ or $(s_t^m, a_t^m, s_{t+1}^m) = (i,j,i')$ |
| $n_t(i) \in [0;M]$ | Number of agents at state $i$ at time $t$ |
| $n_t(i,j) \in [0;M]$ | Number of agents at state $i$ taking action $j$ at time $t$ |
| $n_t(i,j,i') \in [0;M]$ | Number of agents at state $i$ taking action $j$ at time $t$ and transitioning to state $i'$ at time $t+1$ |
| $\mathbf{n}_{\boldsymbol{s}_t}$ | Count table $(n_t(i) \ \forall i \in S)$ |
| $\mathbf{n}_{\boldsymbol{s}_t \boldsymbol{a}_t}$ | Count table $(n_t(i,j) \ \forall i \in S, j \in A)$ |
| $\mathbf{n}_{\boldsymbol{s}_t \boldsymbol{a}_t \boldsymbol{s}_{t+1}}$ | Count table $(n_t(i,j,i') \ \forall i, i' \in S, j \in A)$ |

Table 1: Summary of notations given the state-action trajectories, $(s_{1:H}, a_{1:H})^m \ \forall m$, for all the agents

- We assume a general partially observable setting wherein agents can have different observations based on the collective influence of other agents. An agent observes its local state $s_t^m$. In addition, it also observes $o_t^m$ at time $t$ based on its local state $s_t^m$ and the count table $\mathbf{n}_{\boldsymbol{s}_t}$. E.g., an agent $m$ in state $i$ at time $t$ can observe the count of other agents also in state $i$ ($= n_t(i)$) or other agents in some neighborhood of the state $i$ ($= \{n_t(j) \ \forall j \in \text{Nb}(i)\}$).

- The transition function is $\phi_t\left(s_{t+1}^m = i' | s_t^m = i, a_t^m = j, \mathbf{n}_{\boldsymbol{s}_t}\right)$. The transition function is the same for all the agents. Notice that it is affected by $\mathbf{n}_{\boldsymbol{s}_t}$, which depends on the collective behavior of the agent population.

- Each agent $m$ has a non-stationary policy $\pi_t^m(j|i, o_t^m(i, \mathbf{n}_{\boldsymbol{s}_t}))$ denoting the probability of agent $m$ to take action $j$ given its observation $(i, o_t^m(i, \mathbf{n}_{\boldsymbol{s}_t}))$ at time $t$. We denote the policy over the planning horizon of an agent $m$ to be $\pi^m = (\pi_1^m, \ldots, \pi_H^m)$.

- An agent $m$ receives the reward $r_t^m = r_t(i, j, \mathbf{n}_{\boldsymbol{s}_t})$ dependent on its local state and action, and the counts $\mathbf{n}_{\boldsymbol{s}_t}$.

- Initial state distribution, $b_o = (P(i) \forall i \in S)$, is the same for all agents.

We present here the simplest version where all the agents are of the same type having similar state transition, observation and reward models. The model can handle multiple agent types where agents have different dynamics based on their type. We can also incorporate an *external* state that is unaffected by agents' actions (such as taxi demand in transportation domain). Our results are extendible to address such settings also.

Models such as $\mathbb{C}$Dec-POMDPs are useful in settings where agent population is large, and agent identity does not affect the reward or the transition function. A motivating application of this model is for the taxi-fleet optimization where the problem is to compute policies for taxis such that the total profit of the fleet is maximized (Varakantham et al., 2012; Nguyen et al., 2017). The decision making for a taxi is as follows. At time $t$, each taxi observes its current city zone $z$ (different zones constitute the state-space $S$), and also the count of other taxis in the current zone and its neighboring zones as well as an estimate of the current local demand. This constitutes the count-based observation $o(\cdot)$ for the taxi. Based on this observation, the taxi must decide whether to stay in the current zone $z$ to look for passengers or *move* to another zone. These decision choices depend on several factors such as the ratio of demand and the count of other taxis in the current zone. Similarly, the environment is stochastic with variable taxi demand at different times. Such historical demand data is often available using GPS traces of the taxi fleet (Varakantham et al., 2012).

**Count-Based statistic for planning:** A key property in the $\mathbb{C}$Dec-POMDP model is that the model dynamics depend on the collective interaction among agents rather than agent identities. In settings such as taxi fleet optimization, the agent population size can be quite large ($\approx 8000$ for our real world experiments). Given such a large population, it is not possible to compute unique policy for each agent. Therefore, similar to previous work (Varakantham et al., 2012; Nguyen et al., 2017), our goal is to compute a homogenous policy $\pi$ for all the agents. As the policy $\pi$ is dependent on counts, it represents an expressive class of policies.

For a fixed population $M$, let $\{(s_{1:T}, a_{1:T})^m \ \forall m\}$ denote the state-action trajectories of different agents sampled from the DBN in figure 1. Let $\mathbf{n}_{1:T} = \{(\mathbf{n}_{\boldsymbol{s}_t}, \mathbf{n}_{\boldsymbol{s}_t \boldsymbol{a}_t}, \mathbf{n}_{\boldsymbol{s}_t \boldsymbol{a}_t \boldsymbol{s}_{t+1}}) \ \forall t = 1:T\}$ be the combined vector of the resulting count tables for each time step $t$. Nguyen et al. show that counts $\mathbf{n}$ are the *sufficient statistic* for planning. That is, the joint-value function of a policy $\pi$ over horizon

$H$ can be computed by the expectation over counts as (Nguyen et al., 2017):

$$V(\pi) = \sum_{m=1}^{M} \sum_{T=1}^{H} \mathbb{E}[r_T^m] = \sum_{\mathbf{n} \in \Omega_{1:H}} P(\mathbf{n}; \pi) \left[ \sum_{T=1}^{H} \sum_{i \in S, j \in A} n_T(i,j) r_T(i,j,\mathbf{n}_T) \right] \tag{1}$$

Set $\Omega_{1:H}$ is the set of all allowed consistent count tables as:

$$\sum_{i \in S} n_T(i) = M \ \forall T \ ; \ \sum_{j \in A} n_T(i,j) = n_T(i) \ \forall j \forall T \ ; \ \sum_{i' \in S} n_T(i,j,i') = n_T(i,j) \ \forall i \in S, j \in A, \forall T$$

$P(\mathbf{n}; \pi)$ is the distribution over counts (detailed expression in appendix). A key benefit of this result is that we can evaluate the policy $\pi$ by sampling counts $\mathbf{n}$ directly from $P(\mathbf{n})$ without sampling individual agent trajectories $(s_{1:H}, a_{1:H})^m$ for different agents, resulting in significant computational savings. Our goal is to compute the optimal policy $\pi$ that maximizes $V(\pi)$. We assume a RL setting with centralized learning and decentralized execution. We assume a simulator is available that can provide count samples from $P(\mathbf{n}; \pi)$.

## 3 Policy Gradient for $\mathbb{C}$Dec-POMDPs

Previous work proposed an expectation-maximization (EM) (Dempster et al., 1977) based sampling approach to optimize the policy $\pi$ (Nguyen et al., 2017). The policy is represented as a piecewise linear tabular policy over the space of counts $\mathbf{n}$ where each linear piece specifies a distribution over next actions. However, this tabular representation is limited in its expressive power as the number of pieces is fixed apriori, and the range of each piece has to be defined manually which can adversely affect performance. Furthermore, exponentially many pieces are required when the observation $o$ is multidimensional (i.e., an agent observes counts from some local neighborhood of its location). To address such issues, our goal is to optimize policies in a functional form such as a neural network.

We first extend the policy gradient theorem of (Sutton et al., 1999) to $\mathbb{C}$Dec-POMDPs. Let $\theta$ denote the vector of policy parameters. We next show how to compute $\nabla_\theta V(\pi)$. Let $\boldsymbol{s}_t, \boldsymbol{a}_t$ denote the joint-state and joint-actions of all the agents at time $t$. The value function of a given policy $\pi$ in an expanded form is given as:

$$V_t(\pi) = \sum_{\boldsymbol{s}_t, \boldsymbol{a}_t} P^\pi(\boldsymbol{s}_t, \boldsymbol{a}_t | b_o, \pi) Q_t^\pi(\boldsymbol{s}_t, \boldsymbol{a}_t) \tag{2}$$

where $P^\pi(\boldsymbol{s}_t, \boldsymbol{a}_t | b_o) = \sum_{\boldsymbol{s}_{1:t-1}, \boldsymbol{a}_{1:t-1}} P^\pi(\boldsymbol{s}_{1:t}, \boldsymbol{a}_{1:t} | b_o)$ is the distribution of the joint state-action $\boldsymbol{s}_t, \boldsymbol{a}_t$ under the policy $\pi$. The value function $Q_t^\pi(\boldsymbol{s}_t, \boldsymbol{a}_t)$ is computed as:

$$Q_t^\pi(\boldsymbol{s}_t, \boldsymbol{a}_t) = r_t(\boldsymbol{s}_t, \boldsymbol{a}_t) + \sum_{\boldsymbol{s}_{t+1}, \boldsymbol{a}_{t+1}} P^\pi(\boldsymbol{s}_{t+1}, \boldsymbol{a}_{t+1} | \boldsymbol{s}_t, \boldsymbol{a}_t) Q_{t+1}^\pi(\boldsymbol{s}_{t+1}, \boldsymbol{a}_{t+1}) \tag{3}$$

We next state the policy gradient theorem for $\mathbb{C}$Dec-POMDPs:

**Theorem 1.** *For any $\mathbb{C}$Dec-POMDP, the policy gradient is given as:*

$$\nabla_\theta V_1(\pi) = \sum_{t=1}^{H} E_{\boldsymbol{s}_t, \boldsymbol{a}_t | b_o, \pi} \left[ Q_t^\pi(\boldsymbol{s}_t, \boldsymbol{a}_t) \sum_{i \in S, j \in A} n_t(i,j) \nabla_\theta \log \pi_t \big(j | i, o(i, \mathbf{n}_{\boldsymbol{s}_t}) \big) \right] \tag{4}$$

The proofs of this theorem and other subsequent results are provided in the appendix.

Notice that computing the policy gradient using the above result is not practical for multiple reasons. The space of join-state action $(\boldsymbol{s}_t, \boldsymbol{a}_t)$ is combinatorial. Given that the agent population size can be large, sampling each agent's trajectory is not computationally tractable. To remedy this, we later show how to compute the gradient by directly sampling counts $\mathbf{n} \sim P(\mathbf{n}; \pi)$ similar to policy evaluation in (1). Similarly, one can estimate the action-value function $Q_t^\pi(\boldsymbol{s}_t, \boldsymbol{a}_t)$ using empirical returns as an approximation. This would be the analogue of the standard REINFORCE algorithm (Williams, 1992) for $\mathbb{C}$Dec-POMDPs. It is well known that REINFORCE may learn slowly than other methods that use a learned action-value function (Sutton et al., 1999). Therefore, we next present a function approximator for $Q_t^\pi$, and show the computation of policy gradient by directly sampling counts $\mathbf{n}$.

## 3.1 Policy Gradient with Action-Value Approximation

One can approximate the action-value function $Q_t^\pi(\boldsymbol{s}_t, \boldsymbol{a}_t)$ in several different ways. We consider the following special form of the approximate value function $f_w$:

$$Q_t^\pi(\boldsymbol{s}_t, \boldsymbol{a}_t) \approx f_w(\boldsymbol{s}_t, \boldsymbol{a}_t) = \sum_{m=1}^{M} f_w^m\big(s_t^m, o(s_t^m, \mathbf{n}_{\boldsymbol{s}_t}), a_t^m\big) \qquad (5)$$

where each $f_w^m$ is defined for each agent $m$ and takes as input the agent's local state, action and the observation. Notice that different components $f_w^m$ are correlated as they depend on the common count table $\mathbf{n}_{\boldsymbol{s}_t}$. Such a decomposable form is useful as it leads to efficient policy gradient computation. Furthermore, an important class of approximate value function having this form for $\mathbb{C}$Dec-POMDPs is the *compatible value function* (Sutton et al., 1999) which results in an unbiased policy gradient (details in appendix).

**Proposition 1.** *Compatible value function for $\mathbb{C}$Dec-POMDPs can be factorized as:*

$$f_w(\boldsymbol{s}_t, \boldsymbol{a}_t) = \sum_m f_w^m(s_t^m, o(s_t^m, \boldsymbol{n}_{\boldsymbol{s}_t}), a^m)$$

We can directly replace $Q^\pi(\cdot)$ in policy gradient (4) by the approximate action-value function $f_w$. Empirically, we found that variance using this estimator was high. We exploit the structure of $f_w$ and show further factorization of the policy gradient next which works much better empirically.

**Theorem 2.** *For any value function having the decomposition as:*

$$f_w(\boldsymbol{s}_t, \boldsymbol{a}_t) = \sum_m f_w^m\big(s_t^m, o(s_t^m, \mathbf{n}_{\boldsymbol{s}_t}), a_t^m\big), \qquad (6)$$

*the policy gradient can be computed as*

$$\nabla_\theta V_1(\pi) = \sum_{t=1}^{H} \mathbb{E}_{\boldsymbol{s}_t, \boldsymbol{a}_t} \left[ \sum_m \nabla_\theta \log \pi\big(a_t^m | s_t^m, o(s_t^m, \mathbf{n}_{\boldsymbol{s}_t})\big) f_w^m\big(s_t^m, o(s_t^m, \mathbf{n}_{\boldsymbol{s}_t}), a_t^m\big) \right] \qquad (7)$$

The above result shows that if the approximate value function is factored, then the resulting policy gradient also becomes factored. The above result also applies to agents with multiple types as we assumed the function $f_w^m$ is different for each agent. In the simpler case when all the agents are of same type, then we have the same function $f_w$ for each agent, and also deduce the following:

$$f_w(\boldsymbol{s}_t, \boldsymbol{a}_t) = \sum_{i,j} n_t(i,j) f_w\big(i, j, o(i, \mathbf{n}_{\boldsymbol{s}_t})\big) \qquad (8)$$

Using the above result, we simplify the policy gradient as:

$$\nabla_\theta V_1(\pi) = \sum_t \mathbb{E}_{\boldsymbol{s}_t, \boldsymbol{a}_t} \left[ \sum_{i,j} n_t(i,j) \nabla_\theta \log \pi\big(j | i, o(i, \mathbf{n}_{\boldsymbol{s}_t})\big) f_w(i, j, o(i, \mathbf{n}_{\boldsymbol{s}_t})) \right] \qquad (9)$$

## 3.2 Count-based Policy Gradient Computation

Notice that in (9), the expectation is still w.r.t. joint-states and actions $(\boldsymbol{s}_t, \boldsymbol{a}_t)$ which is not efficient in large population sizes. To address this issue, we exploit the insight that the approximate value function in (8) and the inner expression in (9) depends only on the counts generated by the joint-state and action $(\boldsymbol{s}_t, \boldsymbol{a}_t)$.

**Theorem 3.** *For any value function having the form:* $f_w(\boldsymbol{s}_t, \boldsymbol{a}_t) = \sum_{i,j} n_t(i,j) f_w\big(i, j, o(i, \mathbf{n}_{\boldsymbol{s}_t})\big)$, *the policy gradient can be computed as:*

$$\mathbb{E}_{\mathbf{n}_{1:H} \in \Omega_{1:H}} \left[ \sum_{t=1}^{H} \sum_{i \in S, j \in A} n_t(i,j) \nabla_\theta \log \pi\big(j | i, o(i, \mathbf{n}_t)\big) f_w(i, j, o(i, \mathbf{n}_t)) \right] \qquad (10)$$

The above result shows that the policy gradient can be computed by sampling count table vectors $\mathbf{n}_{1:H}$ from the underlying distribution $P(\cdot)$ analogous to computing the value function of the policy in (1), which is tractable even for large population sizes.

# 4 Training Action-Value Function

In our approach, after count samples $\mathbf{n}_{1:H}$ are generated to compute the policy gradient, we also need to adjust the parameters $w$ of our critic $f_w$. Notice that as per (8), the action value function $f_w(\boldsymbol{s}_t, \boldsymbol{a}_t)$ depends only on the counts generated by the joint-state and action $(\boldsymbol{s}_t, \boldsymbol{a}_t)$. Training $f_w$ can be done by taking a gradient step to minimize the following loss function:

$$\min_w \sum_{\xi=1}^{K} \sum_{t=1}^{H} \left( f_w(\mathbf{n}_t^\xi) - R_t^\xi \right)^2 \tag{11}$$

where $\mathbf{n}_{1:H}^\xi$ is a count sample generated from the distribution $P(\mathbf{n}; \pi)$; $f_w(\mathbf{n}_t^\xi)$ is the action value function and $R_t^\xi$ is the total empirical return for time step $t$ computed using (1):

$$f_w(\mathbf{n}_t^\xi) = \sum_{i,j} n_t^\xi(i,j) f_w(i,j,o(i,\mathbf{n}_t^\xi)); \quad R_t^\xi = \sum_{T=t}^{H} \sum_{i \in S, j \in A} n_T^\xi(i,j) r_T(i,j,\mathbf{n}_T^\xi) \tag{12}$$

However, we found that the loss in (11) did not work well for training the critic $f_w$ for larger problems. Several count samples were required to reliably train $f_w$ which adversely affects scalability for large problems with many agents. It is already known in multiagent RL that algorithms that solely rely on the *global* reward signal (e.g. $R_t^\xi$ in our case) may require several more samples than approaches that take advantage of local reward signals (Bagnell and Ng, 2005). Motivated by this observation, we next develop a local reward signal based strategy to train the critic $f_w$.

**Individual Value Function:** Let $\mathbf{n}_{1:H}^\xi$ be a count sample. Given the count sample $\mathbf{n}_{1:H}^\xi$, let $V_t^\xi(i,j) = \mathbb{E}[\sum_{t'=t}^{H} r_{t'}^m | s_t^m = i, a_m^t = j, n_{1:H}^\xi]$ denote the total expected reward obtained by an agent that is in state $i$ and takes action $j$ at time $t$. This *individual* value function can be computed using dynamic programming as shown in (Nguyen et al., 2017). Based on this value function, we next show an alternative reparameterization of the global empirical reward $R_t^\xi$ in (12):

**Lemma 1.** *The empirical return $R_t^\xi$ for the time step $t$ given the count sample $\mathbf{n}_{1:H}^\xi$ can be reparameterized as: $R_t^\xi = \sum_{i \in S, j \in A} n_t^\xi(i,j) V_t^\xi(i,j)$.*

**Individual Value Function Based Loss:** Given lemma 1, we next derive an upper bound on the on the true loss (11) which effectively utilizes individual value functions:

$$\sum_\xi \sum_t \left( f_w(\boldsymbol{n}^\xi) - R_t^\xi \right)^2 = \sum_\xi \sum_t \left( \sum_{i,j} n_t^\xi(i,j) f_w(i,j,o(i,\mathbf{n}_t^\xi)) - \sum_{i,j} n_t^\xi(i,j) V_t^\xi(i,j) \right)^2$$

$$= \sum_\xi \sum_t \left( \sum_{i,j} n_t^\xi(i,j) \Big( f_w(i,j,o(i,\mathbf{n}_t^\xi)) - V_t^\xi(i,j) \Big) \right)^2 \tag{13}$$

$$\leq M \sum_\xi \sum_{t,i,j} n_t(i,j) \Big( f_w(i,j,o(i,\mathbf{n}_t^\xi)) - V_t^\xi(i,j) \Big)^2 \tag{14}$$

where the last relation is derived by Cauchy-Schwarz inequality. We train the critic using the modified loss function in (14). Empirically, we observed that for larger problems, this new loss function in (14) resulted in much faster convergence than the original loss function in (13). Intuitively, this is because the new loss (14) tries to adjust each critic component $f_w(i,j,o(i,\mathbf{n}_t^\xi))$ closer to its counterpart empirical return $V_t^\xi(i,j)$. However, in the original loss function (13), the focus is on minimizing the global loss, rather than adjusting each individual critic factor $f_w(\cdot)$ towards the corresponding empirical return.

Algorithm 1 shows the outline of our AC approach for $\mathbb{C}$Dec-POMDPs. Lines 7 and 8 show two different options to train the critic. Line 7 represents critic update based on local value functions, also referred to as factored critic update (fC). Line 8 shows update based on global reward or global critic update (C). Line 10 shows the policy gradient computed using theorem 2 (fA). Line 11 shows how the gradient is computed by directly using $f_w$ from eq. (5) in eq. 4.

---
**Algorithm 1:** Actor-Critic RL for ℂDec-POMDPs
---

1   Initialize network parameter $\theta$ for actor $\pi$ and and $w$ for critic $f_w$

2   $\alpha \leftarrow$ actor learning rate

3   $\beta \leftarrow$ critic learning rate

4   **repeat**

5      Sample count vectors $\mathbf{n}_{1:H}^{\xi} \sim P(\boldsymbol{n}_{1:H}; \pi) \; \forall \xi = 1$ to $K$

6      Update critic as:

7      fC : $w = w - \beta \frac{1}{K} \nabla_w \left[ \sum_\xi \sum_{t,i,j} n_t^\xi(i,j)\Big( f_w(i,j,o(i,\mathbf{n}_t^\xi)) - V_t^\xi(i,j)\Big)^2 \right]$

8      C  : $w = w - \beta \frac{1}{K} \nabla_w \left[ \sum_\xi \sum_t \left( \sum_{i,j} n_t^\xi(i,j) f_w(i,j,o(i,\mathbf{n}_t^\xi)) - \sum_{i,j} n_t^\xi(i,j) V_t^\xi(i,j)\right)^2 \right]$

9      Update actor as:

10      fA : $\theta = \theta + \alpha \frac{1}{K} \nabla_\theta \sum_\xi \sum_t \left[ \sum_{i,j} n_t^\xi(i,j) \log \pi \big(j|i,o(i,\mathbf{n}_t^\xi)\big) f_w(i,j,o(\mathbf{n}_t^\xi,i)) \right]$

11      A  : $\theta = \theta + \alpha \frac{1}{K} \nabla_\theta \sum_\xi \sum_t \left[ \sum_{i,j} n_t^\xi(i,j) \log \pi \big(j|i,o(i,\mathbf{n}_t^\xi)\big) \right] \left[ \sum_{i,j} n_t^\xi(i,j) f_w(i,j,o(\mathbf{n}_t^\xi,i)) \right]$

12   **until** *convergence*

13   **return** $\theta, w$

---

## 5   Experiments

This section compares the performance of our AC approach with two other approaches for solving ℂDec-POMDPs—Soft-Max based flow update (SMFU) (Varakantham et al., 2012), and the Expectation-Maximization (EM) approach (Nguyen et al., 2017). SMFU can only optimize policies where an agent's action only depends on its local state, $\pi(a_t^m|s_t^m)$, as it approximates the effect of counts $\mathbf{n}$ by computing the *single* most likely count vector during the planning phase. The EM approach can optimize count-based piecewise linear policies where $\pi_t(a_t^m|s_t^m, \cdot)$ is a piecewise function over the space of all possible count observations $o_t$.

Algorithm 1 shows two ways of updating the critic (in lines 7, 8) and two ways of updating the actor (in lines 10, 11) leading to 4 possible settings for our actor-critic approach—fAfC, AC, AfC, fAC. We also investigate the properties of these different actor-critic approaches. The neural network structure and other experimental settings are provided in the appendix.

For fair comparisons with previous approaches, we use three different models for counts-based observation $o_t$. In 'o0' setting, policies depend only on agent's local state $s_t^m$ and not on counts. In 'o1' setting, policies depend on the local state $s_t^m$ and the single count observation $n_t(s_t^m)$. That is, the agent can only observe the count of other agents in its current state $s_t^m$. In 'oN' setting, the agent observes its local state $s_t^m$ and also the count of other agents from a local neighborhood (defined later) of the state $s_t^m$. The 'oN' observation model provides the most information to an agent. However, it is also much more difficult to optimize as policies have more parameters. The SMFU only works with 'o0' setting; EM and our actor-critic approach work for all the settings.

**Taxi Supply-Demand Matching:**   We test our approach on this real-world domain described in section 2, and introduced in (Varakantham et al., 2012). In this problem, the goal is to compute taxi policies for optimizing the total revenue of the fleet. The data contains GPS traces of taxi movement in a large Asian city over 1 year. We use the observed demand information extracted from this dataset. On an average, there are around 8000 taxis per day (data is not exhaustive over all taxi operators). The city is divided into 81 zones and the plan horizon is 48 half hour intervals over 24 hours. For details about the environment dynamics, we refer to (Varakantham et al., 2012).

Figure 2(a) shows the quality comparisons among different approaches with different observation models ('o0', 'o1' and 'oN'). We test with total number of taxis as 4000 and 8000 to see if taxi population size affects the relative performance of different approaches. The y-axis shows the average per day profit for the entire fleet. For the 'o0' case, all approaches (fAfC-'o0', SMFU, EM-'o0') give similar quality with fAfC-'o0' and EM-'o0' performing slightly better than SMFU for the 8000 taxis. For the 'o1' case, there is sharp improvement in quality by fAfC-'o1' over fAfC-'o0' confirming that taking count based observation into account results in better policies. Our approach fAfC-'o1' is also significantly better than the policies optimized by EM-'o1' for both 4000 and 8000 taxi setting.

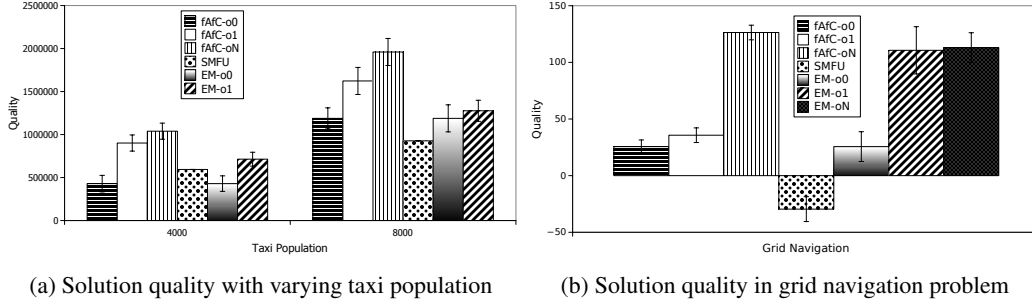

(a) Solution quality with varying taxi population      (b) Solution quality in grid navigation problem

Figure 2: Solution quality comparisons on the taxi problem and the grid navigation

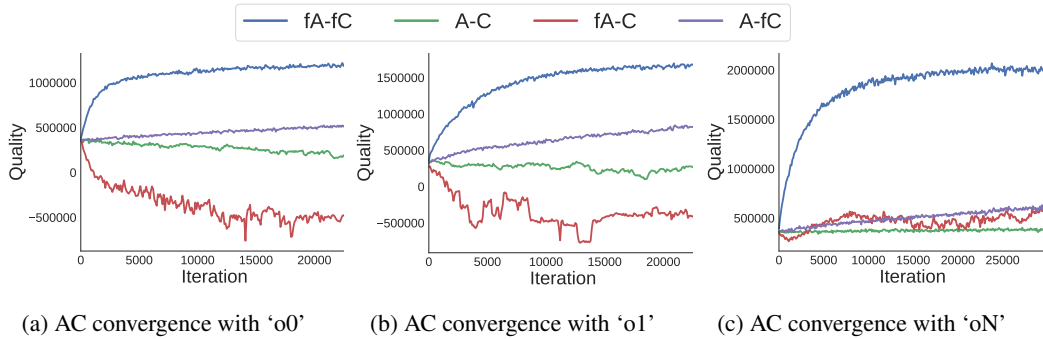

(a) AC convergence with 'o0'     (b) AC convergence with 'o1'     (c) AC convergence with 'oN'

Figure 3: Convergence of different actor-critic variants on the taxi problem with 8000 taxis

To further test the scalability and the ability to optimize complex policies by our approach in the 'oN' setting, we define the neighborhood of each state (which is a zone in the city) to be the set of its geographically connected zones based on the zonal decomposition shown in (Nguyen et al., 2017). On an average, there are about 8 neighboring zones for a given zone, resulting in 9 count based observations available to the agent for taking decisions. Each agent observes both the taxi count and the demand information from such neighboring zones. In figure 2(a), fAfC-'oN' result clearly shows that taking multiple observations into account significantly increases solution quality—fAfC-'oN' provides an increase of 64% in quality over fAfC-'o0' and 20% over fAfC-'o1' for the 8000 taxi case. For EM-'oN', we used a bare minimum of 2 pieces per observation dimension (resulting in $2^9$ pieces per time step). We observed that EM was unable to converge within 30K iterations and provided even worse quality than EM-'o1' at the end. These results show that despite the larger search space, our fAfC approach can effectively optimize complex policies whereas the tabular policy based EM approach was ineffective for this case.

Figures 3(a-c) show the quality Vs. iterations for different variations of our actor critic approach—fAfC, AC, AfC, fAC—for the 'o0', 'o1' and the 'oN' observation model. These figures clearly show that using factored actor and the factored critic update in fAfC is the most reliable strategy over all the other variations and for all the observation models. Variations such as AC and fAC were not able to converge at all despite having exactly the same parameters as fAfC. These results validate different strategies that we have developed in our work to make vanilla AC converge faster for large problems.

**Robot navigation in a congested environment:** We also tested on a synthetic benchmark introduced in (Nguyen et al., 2017). The goal is for a population of robots (= 20) to move from a set of initial locations to a goal state in a 5x5 grid. If there is congestion on an edge, then each agent attempting to cross the edge has higher chance of action failure. Similarly, agents also receive a negative reward if there is edge congestion. On successfully reaching the goal state, agents receive a positive reward and transition back to one of the initial state. We set the horizon to 100 steps.

Figure 2(b) shows the solution quality comparisons among different approaches. In the 'oN' observation model, the agent observes its 4 immediate neighbor node's count information. In this problem, SMFU performed worst, fAfC and EM both performed much better. As expected fAfC-'oN'

provides the best solution quality over all the other approaches. In this domain, EM is competitive with fAfC as for this relatively smaller problem with 25 agents, the space of counts is much smaller than in the taxi domain. Therefore, EM's piecewise policy is able to provide a fine grained approximation over the count range.

# 6 Summary

We addressed the problem of collective multiagent planning where the collective behavior of a population of agents affects the model dynamics. We developed a new actor-critic method for solving such collective planning problems within the $\mathbb{C}$Dec-POMDP framework. We derived several new results for $\mathbb{C}$Dec-POMDPs such as the policy gradient derivation, and the structure of the compatible value function. To overcome the slow convergence of the vanilla actor-critic method we developed multiple techniques based on value function factorization and training the critic using individual value function of agents. Using such techniques, our approach provided significantly better quality than previous approaches, and proved scalable and effective for optimizing policies in a real world taxi supply-demand problem and a synthetic grid navigation problem.

# 7 Acknowledgments

This research project is supported by National Research Foundation Singapore under its Corp Lab @ University scheme and Fujitsu Limited. First author is also supported by A*STAR graduate scholarship.

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
