[Reviews · NeurIPS 2017]

Reviewer 1



** This is an additional review submitted after the rebuttal phase ** This paper presents a policy gradient method for planning in CDec-POMDPs, a useful subclass of DEC-POMDPs where some of the aspects are defined “collectively” instead of “individually”. This collective assumption allows the planning algorithm to scale well to thousands of agents, by reparametrizing the value function (and related terms) in terms of count tables which turns out to be sufficient statistic for planning. The main contribution of this paper is in providing compatible value function theorem for CDec-POMDPs, where it is shown that the compatible value functions are nicely factored due to the collectivity and homogeneous policy assumptions. This work could be a bit incremental extension of Nguyen et al, given that (1) planning via policy gradient by using the planning model as the simulator is a classical idea, and (2) most of the results in this paper heavily depend on the count table reparameterization in the previous work, I think the results presented in this paper is interesting and novel enough to be accepted, although I still have a few reservations. (1) The taxi dispatch domain is not easily accessible, so we cannot truly appreciate how well the algorithm is doing. It would be beneficial to the whole community if the model could be made public. (2) I don’t understand why we need equation (14). I think we can directly optimize equation (13)..?

Reviewer 2



The paper presents a policy gradient algorithm for a multiagent cooperative problem, modeled in a formalism (CDEC-POMDP) whose dynamics, like congestion games, depend on groups of agents rather than individuals. This paper follows the theme of several similar advances in theis field of complex multiagent planning, using factored models to propose practical/tractable approximations. The novelty here is the use of parameterized policies and training algorithms inspired by reinforcement learning (policy gradients). The work is well-motivated, relevant, and particularly well-presented. The theoretical results are new and important. The experiments (on a large taxi domain) are convincing. Overall, the paper advances the state-of-the-art. I have several questions: Q1. What is meant by a "DBN" (... dynamic Bayesian network?) and how does one sample from it? Why is this model shown without even a high-level description? How is it relevant to the rest of the paper? Q2. What is the architecture of the neural net? What is the representation of the input to the network? How large is it (number of parameters, number of layers, units per layer, non-linearities, etc.) Are there two entirely separate networks, one for the critic and one for the policy, or is it a multiheaded network? What optimizer is used, what learning rates, etc. Without these details, these results would not be reproducible. Q3. What is the main motivation for using policy gradient? At first, I thought it would reduce memory as opposed to sample-based tabular value methods. But, it seems like the count tables may dominate the space complexity. Is that true? If not, it would be great to show the reduction in memory by using this method. If so, is the answer then that the benefit of parameterized policy + policy gradient is due to generalization across similar states? Q4. The results show that the algorithm produces better results than the competitor, but it's possible that this setup is particularly more sensitive to parameter values. Even through the authors' claims through the paper, it is implied that the algorithm could be sensitive to the particular choices of training setup. Is the algorithm robust to slight perturbations in values? Comments to improve the paper (does not affect decision): C1. Give some background, even if just a few sentences, on the relationship between policy gradient and actor-critic. This seemed to be assumed, but some readers may not be familiar with this and may be confused without any introduction. (Same goes for DBN, or just drop the term if it's unnecessary.) C2. Somewhere it should be stated that proofs are in the appendix. C3. The min_w should not be in expression (11), sicne the authors just call it a 'loss function', but this is the entire optimization problem. C4. The notation using a superscript xi to represent samples is non-standard and slightly awkward. This xi has no meaning as far as I can tell, so why not use something more familiar like \hat{R}, \hat{\mathbf{n}}, or \tilde{...}?

Reviewer 3



The paper derives a policy gradient method for multi-agent planning. The authors consider a setting with many agents which are treated as a collective population rather than individual agents. The proposed approach looks useful and relevant, although I am not fully convinced by the great novelty of the presented work, as it appears to me as rather straightforward applications of policy gradient to the Collective DEC-POMDP setting described in the paper. The main contribution of the paper seems to be the derivation of the policy gradient for these CDEC-POMDP. This involves the transfer from joint state-action spaces to a count based representation, which was already introduced in the previous work by Nguyen et al. The derivation largely follows the original work by Sutton et al. Since the authors consider a centralized learning, decentralized execution setting, many potential issues of the multi-agent setting are sidestepped. The approach is evaluated on a simulated domain and a large case study based on real data. Several variations of the proposed algorithm are evaluated. Only 1 of proposed variations seems to actually learn to improve performance. The paper cites some recent multi-agent policy gradient papers, but could be better situated with regard to earlier multi-agent policy gradient approaches, e.g. as in Coordinated reinforcement learning (Guestrin et al. 2002) or Learning to cooperate via policy search (Peshkin et al.) , Policy gradient methods for planning (Abderdeen).

Reviewer 4



I am not qualified to review this paper - I am not familiar with the vast work on Dec-POMDPs, let alone CDec-POMDPs. I therefore give my review very low confidence. The paper appears to derive the policy gradient theorem for CDec-POMDPs (I do not know how novel this is, or if people in the multi-agent community find this result interesting). The paper then discusses methods for estimating this gradient. This is challenging due to the combinatorial blow up the the state and action space as the number of agents increases. In the end the authors appear to have a practical first algorithm that can be deployed to benchmark problems (I do not know how standard these benchmarks are, or how difficult they are in comparison to the norm for multi-agent work). There are a few minor typos and grammatical errors, but for the most part the paper is easy to read.